# Self-Supervised Relational Reasoning for Representation Learning

**Massimiliano Patacchiola**
School of Informatics
University of Edinburgh
mpatacch@ed.ac.uk

**Amos Storkey**
School of Informatics
University of Edinburgh
a.storkey@ed.ac.uk

## Abstract

In self-supervised learning, a system is tasked with achieving a surrogate objective by defining alternative targets on a set of unlabeled data. The aim is to build useful representations that can be used in downstream tasks, without costly manual annotation. In this work, we propose a novel self-supervised formulation of relational reasoning that allows a learner to bootstrap a signal from information implicit in unlabeled data. Training a relation head to discriminate how entities relate to themselves (*intra-reasoning*) and other entities (*inter-reasoning*), results in rich and descriptive representations in the underlying neural network backbone, which can be used in downstream tasks such as classification and image retrieval. We evaluate the proposed method following a rigorous experimental procedure, using standard datasets, protocols, and backbones. Self-supervised relational reasoning outperforms the best competitor in all conditions by an average 14% in accuracy, and the most recent state-of-the-art model by 3%. We link the effectiveness of the method to the maximization of a Bernoulli log-likelihood, which can be considered as a proxy for maximizing the mutual information, resulting in a more efficient objective with respect to the commonly used contrastive losses.

## 1   Introduction

Learning useful representations from unlabeled data can substantially reduce dependence on costly manual annotation, which is a major limitation in modern deep learning. Toward this end, one solution is to develop learners able to self-generate a supervisory signal exploiting implicit information, an approach known as self-supervised learning (Schmidhuber, 1987, 1990). Humans and animals are naturally equipped with the ability to learn via an intrinsic signal, but how machines can build similar abilities has been material for debate (Lake et al., 2017). A common approach consists of defining a surrogate task (*pretext*) which can be solved by learning generalizable representations, then use those representations in *downstream* tasks, e.g. classification and image retrieval (Jing and Tian, 2020).

A key factor in self-supervised human learning is the acquisition of new knowledge by relating entities, whose positive effects are well established in studies of adult learning (Gentner and Kurtz, 2005; Goldwater et al., 2018). Developmental studies have shown something similar in children, who can build complex taxonomic names when they have the opportunity to compare objects (Gentner and Namy, 1999; Namy and Gentner, 2002). Comparison allows the learner to neglect irrelevant perceptual features and focus on non-obvious properties. Here, we argue that it is possible to exploit a similar mechanism in self-supervised machine learning via relational reasoning.

The relational reasoning paradigm is based on a key design principle: the use of a relation network as a learnable function to quantify the relationships between a set of *objects*. Starting from this principle, we propose a new formulation of relational reasoning which can be used as a pretext task to build useful representations in a neural network backbone, by training the relation head on unlabeled

data. Differently from the canonical relational approach, which focuses on relations between *objects in the same scene* (Santoro et al., 2017), we focus on relations between *views of the same object* (intra-reasoning) and relations between *different objects in different scenes* (inter-reasoning), in doing so we allow the learner to acquire both intra-class and inter-class knowledge without the need of labeled data.

We evaluate our method following a rigorous experimental methodology, since comparing self-supervised learning methods can be problematic (Kolesnikov et al., 2019; Musgrave et al., 2020). Gains may be largely due to the backbone and learning schedule used, rather than the self-supervised component. To neutralize these effects we provide a benchmark environment where all methods are compared using standard datasets (CIFAR-10, CIFAR-100, CIFAR-100-20, STL-10, tiny-ImageNet, SlimageNet), evaluation protocol (Kolesnikov et al., 2019), learning schedule, and backbones (both shallow and deep). Results show that our method largely outperforms the best competitor in all conditions by an average $14\%$ accuracy and the most recent state-of-the-art method by $3\%$.

*Main contributions*: 1) we propose a novel algorithm based on relational reasoning for the self-supervised learning of visual representations, 2) we show its effectiveness on standard benchmarks with an in-depth experimental analysis, outperforming concurrent state-of-the-art methods (code released with an open-source license[1]), and 3) we highlight how the maximization of a Bernoulli log-likelihood in concert with a relation module, results in more effective and efficient objective functions with respect to the commonly used contrastive losses.

## 1.1 Overview

Following the terminology used in the self-supervised literature (Jing and Tian, 2020) we consider relational reasoning as a *pretext* task for learning useful representations in the underlying neural network backbone. Once the joint system (backbone + relation head) has been trained, the relation head is discarded, and the backbone used in *downstream* tasks (e.g. classification, image retrieval). To achieve this goal we provide a new formulation of relational reasoning. The *canonical formulation* defines it as the process of learning the ways in which entities are connected, using this knowledge to accomplish higher-order goals (Santoro et al., 2017, 2018). The *proposed formulation* defines it as the process of learning the ways entities relate to themselves (intra-reasoning) and to other entities (inter-reasoning), using this knowledge to accomplish downstream goals.

Consider a set of objects $\mathcal{O} = \{o_1, \ldots, o_N\}$, the canonical approach is *within-scene*, meaning that all the elements in $\mathcal{O}$ belong to the same scene (e.g. fruits from a basket). The within-scene approach is not very useful in our case. Ideally, we would like our learner to be able to differentiate between objects taken from every possible scene. Therefore first we define *between-scenes* reasoning: the task of relating objects from different scenes (e.g. fruits from different baskets).

Starting from the between-scenes setting, consider the case where the learner is tasked with discriminating if two objects $\{o_i, o_j\} \sim \mathcal{O}$ belong to the same category $\{o_i, o_j\} \rightarrow$ *same*, or to a different one $\{o_i, o_j\} \rightarrow$ *different*. Often a single attribute is informative enough to solve the task. For instance, in the pair $\{\text{apple}_i, \text{orange}_j\}$ the color alone is a strong predictor of the class, it follows that the learner does not need to pay attention to other features, this results in poor representations.

To solve the issue we alter the object $o_i$ via random augmentations $\mathcal{A}(o_i)$ (e.g. geometric transformation, color distortion) making between-scenes reasoning more complicated. The color of an orange can be randomly changed, or the shape resized, such that it is much more difficult to discriminate it from an apple. In this challenging setting, the learner is forced to take account of the correlation between a wider set of features (e.g. color, size, texture, etc.).

However, it is not possible to create pairs of similar and dissimilar objects when labels are not given. To overcome the problem we bootstrap a supervisory signal directly from the (unlabeled) data, and we do so by introducing *intra-reasoning* and *inter-reasoning*. Intra-reasoning consists of sampling two random augmentations of the same object $\{\mathcal{A}(o_i), \mathcal{A}(o_i)\} \rightarrow$ *same* (positive pair), whereas inter-reasoning consists of coupling two random objects $\{\mathcal{A}(o_i), \mathcal{A}(o_{\setminus i})\} \rightarrow$ *different* (negative pair). This is like coupling different views of the same apple to build the positive pair, and coupling an apple with a random fruit to build the negative pair. In this work we show that it is possible to train a relation module via intra-reasoning and inter-reasoning, with the aim of learning useful representations.

## 2 Previous work

**Relational reasoning.** In the last decades there have been entire sub-fields interested in relational learning: e.g. reinforcement learning (Džeroski et al., 2001) and statistics (Koller et al., 2007). However, only recently the relational paradigm has gained traction in the deep learning community with applications in question answering (Santoro et al., 2017; Raposo et al., 2017), graphs (Battaglia et al., 2018), sequential streams (Santoro et al., 2018), deep reinforcement learning (Zambaldi et al., 2019), few-shot learning (Sung et al., 2018), and object detection (Hu et al., 2018). Our work differentiate from previous one in several ways: (i) previous work is based on labeled data, while we use relational reasoning on unlabeled data; (ii) previous work has focused on within-scene relations, here we focus on relations between different views of the same object (intra-reasoning) and between different objects in different scenes (inter-reasoning); (iii) in previous work training the relation head was the main goal, here is a pretext task for learning useful representations in the underlying backbone.

**Solving pretext tasks.** There has been a substantial effort in defining self-supervised pretext tasks which can be solved only if generalizable representations have been learned. Examples are: predicting the augmentation applied to a patch (Dosovitskiy et al., 2014), predicting the relative location of patches (Doersch et al., 2015), solving Jigsaw puzzles (Noroozi and Favaro, 2016), learning to count (Noroozi et al., 2017), spotting artifacts (Jenni and Favaro, 2018), predicting image rotations (Gidaris et al., 2018), or image channels (Zhang et al., 2017), generating color version of grayscale images (Zhang et al., 2016; Larsson et al., 2016), and generating missing patches (Pathak et al., 2016).

**Metric learning.** The aim of metric learning (Bromley et al., 1994) is to use a distance metric to bring closer representations of similar inputs (positives), while moving away representations of dissimilar inputs (negatives). Commonly used losses are the contrastive loss (Hadsell et al., 2006), the triplet loss (Weinberger et al., 2006), the Noise-Constrative Estimation (NCE, Gutmann and Hyvärinen 2010), the margin (Schroff et al., 2015) and magnet (Rippel et al., 2016) losses. At a first glance relational reasoning and metric learning may seem related, however they are fundamentally different: (i) metric learning explicitly aims at organizing representations by similarity, self-supervised relational reasoning aims at learning a relation measure and, as a byproduct, learning useful representations; (ii) metric learning directly applies a distance metric over the representations, relational reasoning collects representations into a set, aggregates them, then estimates relations; (iii) the relational score is not a distance metric (see Section 3.3) but rather a learnable (probabilistic) similarity measure.

**Contrastive learning.** Metric learning methods based on contrastive loss and NCE are often referred to as contrastive learning methods. Contrastive learning via NCE has recently obtained the state of the art in self-supervised learning. However, one limiting factor is that NCE relies on a large quantity of negatives, which are difficult to obtain in mini-batch stochastic optimization. Recent work has used a memory bank to dynamically store negatives during training (Wu et al., 2018), followed by a plethora of other methods (He et al., 2019; Tian et al., 2019; Misra and van der Maaten, 2019; Zhuang et al., 2019). However, a memory bank has several issues, it introduces additional overhead and a considerable memory footprint. SimCLR (Chen et al., 2020) tries to circumvent the problem by mining negatives in-batch, but this requires specialized optimizers to stabilize the training at scale. We compare relational reasoning and constrastive learning in Section 3.1 and Section 5.

**Pseudo-labeling.** Self-supervision can be achieved providing pseudo-labels to the learner, which are then used for standard supervised learning. A way to obtain pseudo-labels is to use the model itself, picking up the class which has the maximum predicted probability (Lee, 2013; Sohn et al., 2020). A neural network ensemble can also be used to provide the labels (Gupta et al., 2020). In DeepCluster (Caron et al., 2018), pseudo-labels are produced by running a k-means clustering algorithm, which can be forced to induce equipartition (Asano et al., 2020). Recent studies have shown that pseudo-labeling is not competitive against other methods (Oliver et al., 2018), since they are often prone to degenerate solutions with points assigned to the same label (or cluster).

**InfoMax.** A recent line of work has investigated the use of mutual information for unsupervised and self-supervised representation learning, following the InfoMax principle (Linsker, 1988). Mutual information is often maximized at different scales (global and local) on single views (Deep InfoMax, Hjelm et al. 2019), multi-views (Bachman et al., 2019; Ji et al., 2019), or sequentially (Oord et al., 2018). Those methods are often strongly dependent on the choice of feature extractor architecture (Tschannen et al., 2020).

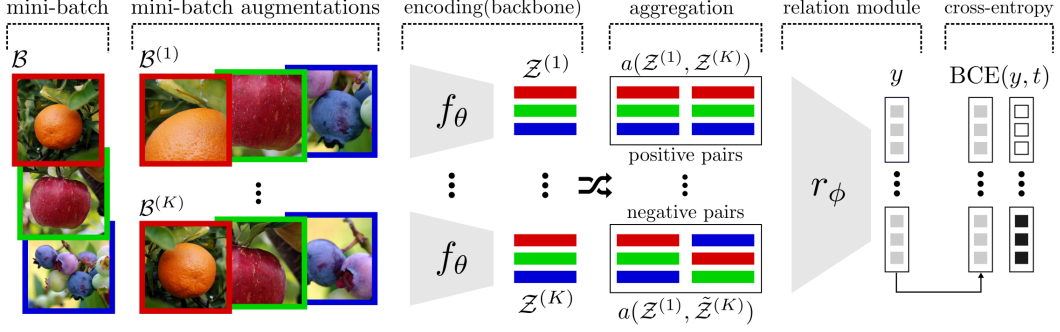

Figure 1: Overview of the proposed method. The mini-batch $\mathcal{B}$ is augmented $K$ times (e.g. via random flip and crop-resize) and passed through a neural network backbone $f_\theta$ to produce the representations $\mathcal{Z}^{(1)}, \ldots, \mathcal{Z}^{(K)}$. An aggregation function $a$ joins positives (representations of the same images) and negatives (randomly paired representations) through a commutative operator. The relation module $r_\phi$ estimates the relational score $y$, which must be 1 for positives and 0 for negatives. The model is optimized minimizing the binary cross-entropy (BCE) between prediction and target $t$.

## 3 Description of the method

Consider an unlabeled dataset $\mathcal{D} = \{\mathbf{x}_n\}_{n=1}^{N}$ and a non-linear function $f_\theta(\cdot)$ parameterized by a vector of learnable weights $\boldsymbol{\theta}$, modeled as a neural network (backbone). A forward pass generates a vector $f_\theta(\mathbf{x}_n) = \mathbf{z}_n$ (representation), which can be collected in a set $\mathcal{Z} = \{\mathbf{z}_n\}_{n=1}^{N}$. The notation $\mathcal{A}(\mathbf{x}_n)$ is used to express the probability distribution of instances generated by applying stochastic data augmentation to $\mathbf{x}_n$, while $\mathbf{x}_n^{(i)} \sim \mathcal{A}(\mathbf{x}_n)$ is the $i$-th sample from this distribution (a particular augmented version of the input instance), and $\mathcal{D}^{(i)} = \{\mathbf{x}_n^{(i)}\}_{n=1}^{N}$ the $i$-th set of random augmentations over all instances. Likewise $\mathbf{z}_n^{(i)} = f_\theta(\mathbf{x}_n^{(i)})$ is grouped in $\mathcal{Z}^{(i)} = \{\mathbf{z}_n^{(i)}\}_{n=1}^{N}$. Let $K$ indicate the total number of augmentations $\mathcal{D}^{(1)}, \ldots, \mathcal{D}^{(K)}$ and their representations $\mathcal{Z}^{(1)}, \ldots, \mathcal{Z}^{(K)}$. Now, let us define a relation module $r_\phi(\cdot)$, as a non-linear function approximator parameterized by $\phi$, which takes as input a pair of aggregated representations and returns a relation score $y$. Indicating with $a(\cdot, \cdot)$ an aggregation function and with $\mathcal{L}(y, t)$ the loss between the score and a target value $t$, the complete learning objective can be specified as

$$\underset{\boldsymbol{\theta}, \boldsymbol{\phi}}{\operatorname{argmin}} \sum_{n=1}^{N} \sum_{i=1}^{K} \sum_{j=1}^{K} \underbrace{\mathcal{L}\Big(r_\phi\big(a(\mathbf{z}_n^{(i)}, \mathbf{z}_n^{(j)})\big), t=1\Big)}_{\text{intra-reasoning}} + \underbrace{\mathcal{L}\Big(r_\phi\big(a(\mathbf{z}_n^{(i)}, \mathbf{z}_{\backslash n}^{(j)})\big), t=0\Big)}_{\text{inter-reasoning}}, \text{ with } \mathbf{z}_n = f_\theta(\mathbf{x}_n),$$

(1)

where $\backslash n$ is an index randomly sampled from $\{1, \ldots, N\} \setminus \{n\}$. In practice (1) can be framed as a standard binary classification problem (see Section 3.4), and minimized by stochastic gradient descent sampling a mini-batch $\mathcal{B} \sim \mathcal{D}$ with pairs built by repeatedly applying $K$ augmentations to $\mathcal{B}$. Positives can be obtained pairing two encodings of the same input (intra-reasoning term), and negatives by randomly coupling representations of different inputs (inter-reasoning term), relying on the assumption that in common settings this yields a very low probability of false negatives. An overview of the model is given in Figure 1 and the pseudo-code in Appendix C (supp. material).

**Mutual information.** Following the recent work of Boudiaf et al. (2020) we can interpret (1) in terms of *mutual information*. Let us define the random variables $Z|X$ and $T|Z$, representing embeddings and targets. Now consider the generative view of mutual information

$$I(Z; T) = H(Z) - H(Z|T).$$

(2)

Intra-reasoning is a tightening factor which can be expressed as a bound over the conditional entropy $H(Z|T)$. Inter-reasoning is a scattering factor which can be linked to the entropy of the representations $H(Z)$. In other words, each representation is pushed towards a positive neighborhood (intra-reasoning) and repelled from a complementary set of negatives (inter-reasoning). Under this interpretation (1) can be considered as a proxy for maximizing Equation (2). We refer the reader to Boudiaf et al. (2020) for a more detailed analysis.

## 3.1 Inputs augmentation

Given a random mini-batch of $M$ input instances $\mathcal{B} \sim \mathcal{D}$, recursively apply data augmentation $K$ times $\mathcal{B}^{(1)}, \ldots, \mathcal{B}^{(K)}$ then propagate through $f_\theta$ with a forward pass, to generate the corresponding representations $\mathcal{Z}^{(1)}, \ldots, \mathcal{Z}^{(K)}$. Representations are coupled across augmentations to generate positive and negative tuples

$$\forall i, j \in \{1, \ldots, K\} \quad \big( \underbrace{\mathcal{Z}^{(i)}, \mathcal{Z}^{(j)}}_{\text{positives}} \big) \quad \text{and} \quad \big( \underbrace{\mathcal{Z}^{(i)}, \tilde{\mathcal{Z}}^{(j)}}_{\text{negatives}} \big), \tag{3}$$

where $\tilde{\mathcal{Z}}$ indicates random assignment of each representation $\mathbf{z}_n^{(i)}$ to a different element $\mathbf{z}_{\backslash n}^{(j)}$. In practice, we discard identical pairs (identity mapping is learned across augmentations) and take just one of the symmetrical tuples $(\mathbf{z}^{(i)}, \mathbf{z}^{(j)})$ and $(\mathbf{z}^{(j)}, \mathbf{z}^{(i)})$ (the aggregation function ensures commutation, see Section 3.2). If a certain amount of data in $\mathcal{D}$ is labeled (semi-supervised setting), then positive pairs include representations of different augmented inputs belonging to the same category.

**Computational cost.** Having defined $M$ as the number of inputs in the mini-batch $\mathcal{B}$, and $K$ as the number of augmentations, the total number of pairs $P$ (positive and negative) is given by

$$P = M(K^2 - K). \tag{4}$$

The number of comparisons $P$ scales quadratically with the number of augmentations $K$, and linearly with the size of the mini-batch $M$; whereas in recent constrastive learning methods (Chen et al., 2020), they scale as $P = (MK)^2$, which is quadratic in both augmentations and mini-batch size.

**Augmentation strategy.** Here, we consider the particular case where the input instances are color images. Following previous work (Chen et al., 2020) we focus on two augmentations: random crop-resize and color distortion. Crop-resize enforces comparisons between views: global-to-global, global-to-local, and local-to-local. Since augmentations are sampled from the same color distribution, the color alone may suffice to distinguish positives and negatives. Color distortion enforces color-invariant encodings and neutralizes learning shortcuts. Additional details about the augmentations used in this work are reported in Section 4 and Appendix A.3 (supp. material).

## 3.2 Aggregation function

Relation networks operate over sets. To avoid a combinatorial explosion due to an increasing cardinality, a commutative aggregation function is applied. Given $f_\theta(\mathbf{x}_i) = \mathbf{z}_i$ and $f_\theta(\mathbf{x}_j) = \mathbf{z}_j$, there are different possible choices for the aggregation function

$$a_{\text{sum}}(\mathbf{z}_i, \mathbf{z}_j) = \mathbf{z}_i + \mathbf{z}_j, \quad a_{\text{max}}(\mathbf{z}_i, \mathbf{z}_j) = \max(\mathbf{z}_i, \mathbf{z}_j), \quad a_{\text{cat}}(\mathbf{z}_i, \mathbf{z}_j) = (\mathbf{z}_i, \mathbf{z}_j), \tag{5}$$

where sum and max are applied elementwise. Concatenation $a_{\text{cat}}$ is not commutative, but it has been previously used when the cardinality is small (Hu et al., 2018; Sung et al., 2018), like in our case.

## 3.3 Relation module

The relation module is a function $r_\phi(\cdot)$ parameterized by a vector of learnable weights $\phi$, modeled as a multi-layer perceptron (MLP). Given a pair of representations $\mathbf{z}_i$ and $\mathbf{z}_j$, the module takes as input the aggregated pair and produce a scalar $y$ (relation score)

$$r_\phi\big(a(\mathbf{z}_i, \mathbf{z}_j)\big) = y. \tag{6}$$

The relational score respects two properties: (i) $r(a(\mathbf{z}_i, \mathbf{z}_j)) \in [0, 1]$; (ii) $r(a(\mathbf{z}_i, \mathbf{z}_j)) = r(a(\mathbf{z}_j, \mathbf{z}_i))$. It is crucial to not misinterpret the relational score for a pairwise distance metric. Given a set of input vectors $\{\mathbf{v}_i, \mathbf{v}_j, \mathbf{v}_k\}$ the distance metric $d(\cdot, \cdot)$ respects four properties: (i) $d(\mathbf{v}_i, \mathbf{v}_j) \geq 0$; (ii) $d(\mathbf{v}_i, \mathbf{v}_j) = 0 \leftrightarrow \mathbf{v}_i = \mathbf{v}_j$; (iii) $d(\mathbf{v}_i, \mathbf{v}_j) = d(\mathbf{v}_j, \mathbf{v}_i)$; (iv) $d(\mathbf{v}_i, \mathbf{v}_k) \leq d(\mathbf{v}_i, \mathbf{v}_j) + d(\mathbf{v}_j, \mathbf{v}_k)$. Note that the relational score does not satisfies all the conditions of a distance metric and therefore *the relational score is not a distance metric*, but rather a probabilistic estimate (see Section 3.4).

## 3.4 Definition of the loss

The learning objective (1) can be framed as a binary classification problem over the $P$ representation pairs. Under this interpretation, the relation score $y$ represents a probabilistic estimate of representation membership, which can be induced through a sigmoid activation function. It follows that the objective reduces to the maximization of a Bernoulli log-likelihood, or similarly, the minimization of a binary cross-entropy loss

$$\mathcal{L}(\mathbf{y}, \mathbf{t}, \gamma) = \frac{1}{P} \sum_{i=1}^{P} -w_i \Big[ t_i \cdot \log y_i + (1 - t_i) \cdot \log(1 - y_i) \Big], \tag{7}$$

with target $t_i = 1$ for positives and $t_i = 0$ for negatives. The optional weight $w_i$ is a scaling factor

$$w_i = \frac{1}{2} \Big[ (1 - t_i) \cdot y_i + t_i \cdot (1 - y_i) \Big]^{\gamma}, \tag{8}$$

where $\gamma \geq 0$ defines how sharp the weight should be. This factor gives more importance to uncertain estimations and it is also known as the focal loss (Lin et al., 2017).

## 4 Experiments

Evaluating self-supervised methods is problematic because of substantial inconsistency in the way methods have been compared (Kolesnikov et al., 2019; Musgrave et al., 2020). We provide a standardized environment implemented in Pytorch (see code in supp. material) using standard datasets (CIFAR-10, CIFAR-100, CIFAR-100-20, STL-10, tiny-ImageNet, SlimageNet), different backbones (shallow and deep), same learning schedule (epochs), and well know evaluation protocols (Kolesnikov et al., 2019). In most conditions our method show superior performance.

**Implementation.** Hyperparameters (relation learner): mini-batch of 64 images ($K = 16$ for ResNet-32 on tiny-ImageNet, $K = 25$ for ResNet-34 on STL-10, $K = 32$ for the rest), Adam optimizer with learning rate $10^{-3}$, binary cross-entropy loss with focal factor ($\gamma = 2$). Relation module: MLP with 256 hidden units (batch-norm + leaky-ReLU) and a single output unit (sigmoid). Aggregation: we used concatenation as it showed to be more effective (see Appenidx B.8, Table 13 supp. material). Augmentations: horizontal flip (50% chance), random crop-resize, conversion to grayscale (20% chance), and color jitter (80% chance). Backbones: Conv-4, ResNet-8/32/56 and ResNet-34 (He et al., 2016). Baselines: DeepCluster (Caron et al., 2018), RotationNet (Gidaris et al., 2018), Deep InfoMax (Hjelm et al., 2019), and SimCLR (Chen et al., 2020). Those are recent (hard) baselines, with SimCLR being the current state-of-the-art in self-supervised learning. As upper bound we include the performance of a fully supervised learner (it has access to the labels), and as lower bound a network initialized with random weights, evaluated training only the linear classifier. All results are the average over three random seeds. Additional details in supp. material (Appendix A).

**Linear evaluation.** We follow the linear evaluation protocol defined by Kolesnikov et al. (2019) training the backbone for 200 epochs using the *unlabeled* training set, and then training for 100 epochs a linear classifier on top of the backbone features (without backpropagation in the backbone weights). The accuracy of this classifier on the test set is considered as the final metric to asses the quality of the representations. Our method largely outperforms other baselines with an accuracy of 46.2% (CIFAR-100) and 30.5% (tiny-Imagenet), which is an improvement of +4.0% and +4.7% over the best competitor (SimCLR), see Table 1. Best results are also obtained with the Conv-4 backbone on all datasets. Only in CIFAR-10/ResNet-32 SimCLR is doing better, with a score of 77% against 75% of our method, see Appendix B.1 (supp. material). In the appendix we report the results on the challenging SlimageNet dataset used in few-shot learning (Antoniou et al., 2020): 160 low-resolution images for each one of the 1000 classes in ImageNet. On SlimageNet our method has the highest accuracy (15.8%, $K = 16$), being better than RotationNet (7.2%) and SimCLR (14.3%).

**Domain transfer.** We evaluate the performance of all methods in transfer learning by training on the unlabeled CIFAR-10 with linear evaluation on the labeled CIFAR-100 (and viceversa). Our method outperforms once again all the others in every condition. In particular, it is very effective in generalizing from a simple dataset (CIFAR-10) to a complex one (CIFAR-100), obtaining an accuracy of 41.5%, which is a gain of +5.3% over SimCLR and +7.5% over the supervised baseline (with linear transfer). For results see Table 1 and Appendix B.2 (supp. material).

Table 1: Comparison on various benchmarks. Mean accuracy (percentage) and standard deviation over three runs (ResNet-32). Best results in bold. **Linear Evaluation:** training on unlabeled data and linear evaluation on labeled data. **Domain Transfer:** training on unlabeled CIFAR-10 and linear evaluation on labeled CIFAR-100 (10→100), and viceversa (100→10). **Grain:** training on unlabeled CIFAR-100, linear evaluation on coarse-grained CIFAR-100-20 (20 super-classes). **Finetune:** training on the unlabeled set of STL-10, finetuning on the labeled set (ResNet-34).

| | Linear Evaluation | | Domain Transfer | | Grain | Finetune |
|---|---|---|---|---|---|---|
| **Method** | **CIFAR-100** | **tiny-ImgNet** | **10→100** | **100→10** | **CIFAR-100-20** | **STL-10** |
| Supervised (upper bound) | 65.32±0.22 | 50.09±0.32 | 33.98±0.71 | 71.01±0.44 | 76.35±0.57 | 69.82±3.36 |
| Random Weights (lower bound) | 7.65±0.44 | 3.24±0.43 | 7.65±0.44 | 27.47±0.83 | 16.56±0.48 | n/a |
| DeepCluster (Caron et al., 2018) | 20.44±0.80 | 11.64±0.21 | 18.37±0.41 | 43.39±1.84 | 29.49±1.36 | 73.37±0.55 |
| RotationNet (Gidaris et al., 2018) | 29.02±0.18 | 14.73±0.48 | 27.02±0.20 | 52.22±0.70 | 40.45±0.39 | 83.29±0.44 |
| Deep InfoMax (Hjelm et al., 2019) | 24.07±0.05 | 17.51±0.15 | 23.73±0.04 | 45.05±0.24 | 33.92±0.34 | 76.03±0.37 |
| SimCLR (Chen et al., 2020) | 42.13±0.35 | 25.79±0.35 | 36.20±0.16 | 65.59±0.76 | 51.88±0.48 | 89.31±0.14 |
| *Relational Reasoning* (ours) | **46.17±0.17** | **30.54±0.42** | **41.50±0.35** | 67.81±0.42 | **52.44±0.47** | **89.67±0.33** |

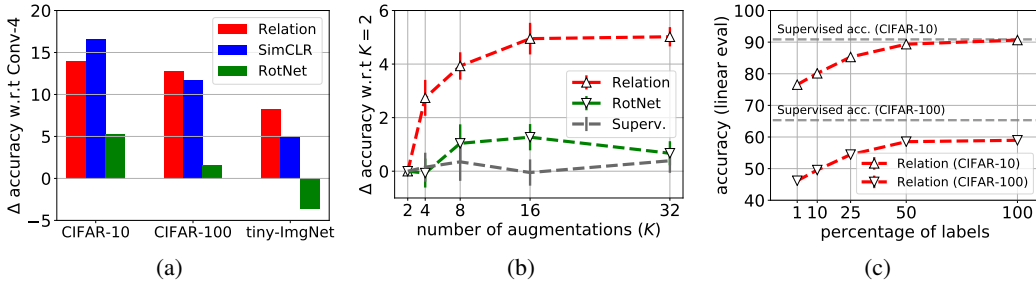

&emsp;&emsp;&emsp;&emsp;(a)&emsp;&emsp;&emsp;&emsp;&emsp;&emsp;&emsp;&emsp;&emsp;&emsp;(b)&emsp;&emsp;&emsp;&emsp;&emsp;&emsp;&emsp;&emsp;&emsp;&emsp;(c)

Figure 2: (a) Difference in accuracy using the deeper backbone (Conv4→ResNet-32, linear evaluation). As the complexity of the dataset raises our method performs increasingly better than the others. (b) Correlation between validation accuracy (3 seeds, Conv-4, CIFAR-10) and number of mini-batch augmentations. Only in our method the accuracy is positively correlated with the number of augmentations. (c) Semi-supervised accuracy with an increasing percentage of labels (ResNet-32).

**Grain.** Different methods produce different representations, some may be better on datasets with a small amount of labels (coarse-grained), others may be better on datasets with a large amount of labels (fine-grained). To investigate the granularity of the representations we train on unlabeled CIFAR-100, then perform linear evaluation using the 100 labels (fine grained; e.g. apple, fox, bee, etc) and the 20 super-labels (coarse grained; e.g. fruits, mammals, insects, etc). Also in this case our method is superior in all conditions with an accuracy of 52.4% on CIFAR-100-20, see Table 1 and Appendix B.3 (supp. material). In comparison, the method does better in the fine-grained case, indicating that it is well suited for datasets with a large amount of classes.

**Finetuning.** We used the STL-10 dataset (Coates et al., 2011) which provides a set of unlabeled data coming from a similar but different distribution from the labeled data. Methods have been trained for 300 epochs on the unlabeled set (100K images), finetuned for 20 epochs on the labeled set (5K images), and finally evaluated on the test set (8K images). We used a mini-batch of 64 with $K = 25$ and a ResNet-34. Implementation details are reported in Appendix A.6 (supp. material). Results in Table 1 show that our method obtains the highest accuracy: 89.67% (best seed 90.04%). Moreover a wider comparison reported in Appendix B.4 (supp. material) shows that the method outperforms strong supervised baselines and the previous self-supervised state-of-the-art (88.80%, Ji et al., 2019).

**Depth of the backbone.** In Appendix B.5 we report an extensive comparison on four backbones of increasing depth: Conv-4, ResNet-8, ResNet-32, and ResNet-56. We tested the three best methods (RotationNet, SimCLR, and Relational Reasoning) on CIFAR-10/100 linear evaluation, grain, and domain transfer for a total of 24 conditions. Results show that our method has the highest accuracy on 21 of those conditions, with SimCLR performing better on CIFAR-10 linear evaluation with ResNet backbones. A distilled version of those results is reported in Figure 2a. The figure shows the gain in accuracy from using a ResNet-32 instead of a Conv-4 backbone for datasets of increasing complexity (10, 100, and 200 classes). As the complexity of the dataset raises our method performs increasingly

better than the others. The relative gain against SimCLR gets larger: $-2.6\%$ (CIFAR-10), $+1.1\%$ (CIFAR-100), $+3.3\%$ (tiny-ImageNet). The relative gain against RotationNet is even more evident: $+8.7\%$, $+11.2\%$, $+11.9\%$.

**Additional experiments.** Figure 2b and Appendix B.6 (supp. material) show the difference in accuracy between $K = 2$ and $K \in \{4, 8, 16, 32\}$ mini-batch augmentations for a fixed mini-batch size. There is a clear positive correlation between the number of augmentations and the performance of our model, while the same does not hold for a self-supervised algorithm (RotationNet) and the supervised baseline. Figure 2c and Appendix B.7 (supp. material) show the accuracy obtained via linear evaluation when the number of available labels is gradually increased (0%, 1%, 10%, 25%, 50%, 100%), in both CIFAR-10 and CIFAR-100 (ResNet-32). The accuracy is positively correlated with the proportion of labels available, approaching the supervised upper bound when 100% of labels are available.

**Ablations.** In Appendix B.8 we report the results of ablation studies on the aggregation function and relation head. We compare four aggregation functions: sum, mean, maximum, and concatenation. Results show that concatenation and maximum are respectively the most and less effective functions. Concatenation may favor backpropagation improving the quality of the representations, as supported by similar results in previous work (Sung et al., 2018). Ablations of the relation head have followed two directions: (i) removing the head, and (ii) replacing the relation module with an encoder. In the first condition we removed the head and replace it with a simple dot product between representation pairs (BCE-focal loss). In the second condition we followed an approach similar to SimCLR (Chen et al., 2020), replacing the relation head with an encoder and applying the dot product to representations at the higher level (BCE-focal loss). The second condition differs from SimCLR for the loss type (BCE vs Contrastive) and total number of mini-batch augmentations ($K = 32$ vs $K = 2$). In both conditions we observe a severe degradation of the performance with respect to the complete model (from a minimum of $-3\%$ to a maximum of $-23\%$), confirming that the relation module is a fundamental component in the pipeline (see discussion in Section 5).

**Qualitative analysis.** In Appendix B.9 (supp. material) is presented a qualitative comparison between the proposed method and RotationNet, on an image retrieval downstream task. Given a random query image (not cherry-picked) the top-10 most similar images in representation space are retrieved. Our method shows better distinction between categories which are hard to separate (e.g. ships vs planes, trucks vs cars). The lower sample variance and the higher similarity with the query, confirm the fine-grained organization of the representations, which account for color, texture, and geometry. An analysis of retrieval errors in Appendix B.10 (supp. material) shows that the proposed method is superior in accuracy across all categories while being more robust against misclassification, with a top-10 retrieval accuracy of 67.8% against 47.7% of RotationNet. In Appendix B.11 (supp. material) we report a qualitative analysis of the representations (ResNet-32, CIFAR-10) using t-SNE (Maaten and Hinton, 2008). Relational reasoning is able to aggregate the data in a more effective way, and to better capture high level relations with lower scattering (e.g. vehicles vs animals super-categories).

## 5    Discussion and conclusions

Self-supervised relational reasoning is effective on a wide range of tasks in both a quantitative and qualitative manner, and with backbones of different size (ResNet-32, ResNet-56 and ResNet-34, with $0.5 \times 10^6$, $0.9 \times 10^6$ and $21.3 \times 10^6$ parameters). Representations learned through comparison can be easily transferred across domains, they are fine-grained and compact, which may be due to the direct correlation between accuracy and number of augmentations. An instance is pushed towards a positive neighborhood (intra-reasoning) and repelled from a complementary set of negatives (inter-reasoning). The number of augmentations may have a primary role in this process affecting the quality of the clusters. The possibility to exploit an high number of augmentations, by generating them on the fly, could be decisive in the low-data regime (e.g. unsupervised few-shot/online learning) where self-supervised relational reasoning has the potential to thrive. Those are factors that require further consideration and investigation.

**From self-supervised to supervised.** Recent work has showed that contrastive learning can be used in a supervised setting with competitive results (Khosla et al., 2020). In our experiments we have observed a similar trend, with relational reasoning approaching the supervised performance when all the labels are available. However, we have obtained those results using the same hyperparameters and

augmentations used in the self-supervised case, while there may be alternatives that are more effective. Learning by comparison could help in disentangling fine-grained differences in a fully supervised setting with high number of classes, and be decisive to build complex taxonomic representations, as pointed out in cognitive studies (Gentner and Namy, 1999; Namy and Gentner, 2002)

**Comparison with contrastive methods.** We have compared relational reasoning to a state-of-the-art contrastive learning method (SimCLR) using the same backbone, head, augmentation strategy, and learning schedule. Relational reasoning outperforms SimCLR (+3% on average) using a lower number of pairs, being more efficient. Given a mini-batch of size 64, relational reasoning uses $6.35 \times 10^4$ ($K = 32$) and $1.5 \times 10^4$ ($K = 16$) pairs, against $6.55 \times 10^4$ of SimCLR with mini-batch 128. Contrastive losses needs a large number of negatives, which can be gathered by increasing $M$ the size of the mini-batch, or increasing $K$ the number of augmentations (both solutions incur a quadratic cost, see Section 3.1). High quality negatives can only be gathered following the first solution, since the second provides lower sample variance. A typical mini-batch in SimCLR encloses 98% negatives and 2% positives, in our method 50% negatives and 50% positives. The larger set of positives could be one of the reasons why relational reasoning is more effective in disentangling fine-grained representations. In addition to the difference in loss type, there is an important structural difference between the two approaches: in SimCLR pairs are allocated in the loss space and then compared via dot product, while in relational reasoning they are aggregated in the space of transferable representations and compared through a relation head. Ablation studies in Section 4 have shown that this structural difference is fundamental for obtaining higher performances, but the way it influences the learning dynamics and the optimization process is not clear and requires further investigation.

**Why does cross-entropy work so well?** We argue that in the context of recent state-of-the-art methods, cross-entropy has been overlooked in favor of contrastive losses. Our experiments show that cross-entropy is a more efficient and effective objective function with respect to the commonly used contrastive losses. Based on the results of the ablation studies, we hypothesize that the difference in performance is mainly due to the use of a relation module in conjunction with the binary cross-entropy loss. When the BCE is split from the relation head and applied directly to the representations there is a drastic drop in performance; applying the BCE to surrogate representations in a second encoding stage (like in SimCLR) is equally ineffective. Therefore, the use of BCE on its own does not provide any advantage but in concert with the relation head it becomes effective. A more thorough analysis is necessary to substantiate these findings, which is left for future work.

## Broader Impact

The motivation behind this work is to build systems able to exploit a large amount of unlabeled data. Applications that could benefit from the proposed method span from standard supervised classifiers to medical diagnostic systems. Therefore, there is a large number of individuals who may benefit or be harmed from this research. This requires putting some effort into selecting the data source, especially when the system is scaled.

In most cases a large body of unlabeled images can be easily gathered from the internet; to avoid biases those images should be representative of different categories. Our method does not guarantee unbiased predictions, therefore it should be used with caution in critical applications. Individuals who may want to use it should consider the particular source of data at hand and evaluate how it could impact the system performance after the final deployment.

## Acknowledgments and Disclosure of Funding

This work was supported by a Huawei DDMPLab Innovation Research Grant.

MP and AS would like to thank anonymous reviewers for useful comments and suggestions; the BayesWatch team for feedback and discussion, in particular Elliot J. Crowley, Luke Darlow, and Joseph Mellor. MP would like to thank the Becchi team for revising the preliminary version of the manuscript, in particular Valerio Biscione, Riccardo Polvara, and Luca Surace.

## Footnotes

[1] https://github.com/mpatacchiola/self-supervised-relational-reasoning

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
