[Supplementary Material]

# A  Implementation details

## A.1  Datasets

For datasets with low/medium number of categories we used CIFAR-10 and CIFAR-100 (Krizhevsky et al., 2009), which are composed of $32 \times 32$ RGB images, with 10 and 100 classes respectively. In addition we used the 20 super-classes of CIFAR-100 (naming this CIFAR-100-20), which consists of broader categories (e.g. fruits, mammals, insects, etc). In the finetuning experiments we used the STL-10 dataset (Coates et al., 2011) which provides 100K RGB images of size $96 \times 96$ in the unlabeled set, 5K images in the labeled set, and 8K images in the test set.

For datasets with an high number of categories we used the tiny-ImageNet and SlimageNet (Antoniou et al., 2020) datasets, both of them derived from ImageNet (Russakovsky et al., 2015). Tiny-ImageNet consists of 200 different categories, with 500 training images ($64 \times 64$, 100K in total), 50 validation images (10K in total), and 50 test images (10K in total). SlimageNet consists of $64 \times 64$ RGB images, 1000 categories with 160 training images (160K in total), 20 validation images (20K in total), and 20 test images (20K in total). Both of them are considered more challenging than ImageNet because of the lower resolution of the images and lower number of training samples.

## A.2  Backbones

We use off-the-shelf Pytorch implementations of ResNets as described in the original paper (He et al., 2016). Some of these networks have quite different structure, with ResNet-8/32/56 based on three hyper-blocks (ResNet-32 has $0.5 \times 10^6$ total parameters) and ResNet-34 based on four hyper-blocks ($21.3 \times 10^6$ total parameters). The Conv-4 backbone is based on three blocks (8, 16, 32 feature maps), each one performing: convolution (kerne-size=3, stride=1, padding=1), BatchNorm, ReLU, average pooling (kerne-size=2, stride=2). The fourth block (64 feature maps) performed the same operations but with an adaptive average pooling to squeeze the maps to unit shape in the spatial dimension. We used standard fan-in/fan-out weight initialization, and set BatchNorm weights to 1 and bias to 0. For Conv-4 and ResNet-8/32/56 the size of the representations is 64, whereas for ResNet-34 is 512.

## A.3  Augmentations

During the self-supervised training phase of our method we used a set of augmentations which is similar to the one adopted by Chen et al. (2020). We apply horizontal flip (50% chance), random crop-resize, conversion to grayscale (20% chance), and color jitter (80% chance). Random crop-resize consists of cropping the given image (from 0.08 to 1.0 of the original size), changing the aspect ratio (from 3/4 to 4/3 of the original aspect ratio), and finally resizing to input shape using a bilinear interpolation. Color jitter consists of sampling from a uniform distribution $[0, max]$ a jittering value for: brightness ($max = 0.8$), contrast ($max = 0.8$), saturation ($max = 0.8$), and hue ($max = 0.2$).

## A.4  Computing infrastructure

All the experiments have been performed on a workstation with 20 cores, 187 GB of RAM, and with 8 NVIDIA GeForce RTX 2080 Ti GPUs (11 GB of internal RAM). All the methods could fit on a single one of those GPUs.

## A.5  Other methods

**Supervised.** This baseline consists of standard supervised training. We used standard data augmentation (horizontal flip and random crop) and learning schedule (SGD optimizer with initial learning rate of 0.1 divided by 10 at 50% and 75% of total epochs). It represents an upper bound. When evaluated for the number of augmentations (Appendix B.6) the same strategy adopted in our method (Appendix A.3) has been used to augment the input mini-batch (size 128) $K$ times with coherent labels.

**Random weights.** This baseline consists of initializing the weights of the backbone via standard fan-in/fan-out, then perform linear evaluation optimizing the last linear layer (without backpropagation on the backbone). It represents a lower bound since the backbone is not trained.

**DeepCluster (Caron et al., 2018).** We adapted the open-source implementation provided by the authors[2]. Clustering has been performed at the beginning of each epoch by using the k-means algorithm available in Scikit-learn. We performed whitening over the features before the clustering step, as suggested by the authors. We used a number of cluster one order of magnitude larger than the number of classes in the dataset, as recommended by the authors to improve the performance. We also used an MLP head (256 hidden units with leaky-ReLU

and BatchNorm) instead of a linear layer, since in our tests this showed to slightly boost the performance. The MLP weights have been reset at the beginning of each epoch as in the original code. We optimized the model minimizing the cross-entropy loss between the pseudo-labels provided by the clustering and the network outputs. We used Adam optimizer with learning rate $10^{-3}$.

**RotationNet (Gidaris et al., 2018).** Given the simplicity of the method, this has been reproduced locally following the instructions of the authors. Labels are provided by 4 rotations ($0°, 90°, 180°, 270°$), those are the one providing the highest accuracy according to the authors. The input mini-batch of size 128, has been augmented adding 4 rotations for each image (resulting in a mini-batch of size $128 \times 4$). This is in line with the best performing strategy reported by the authors. In all experiments the cross-entropy loss between the network output and the labels provided by the rotation has been minimized (Adam optimizer, learning rate $10^{-3}$). When evaluated for the number of augmentations (Appendix B.6) the same strategy used in our method has been applied (Appendix A.3), augmenting the input mini-batch (size 128) $K$ times with coherent self-supervised rotation labels. In order to keep the size of the mini-batch manageable the additional 4 rotations for image have not been included, since this would increase the size to $4 \times K$ and not fit on the available hardware.

**Deep InfoMax (Hjelm et al., 2019)** The code has been adapted from open-source implementations available online (see code for details) and from the code provided by the authors[3]. The local version of the algorithm has been used ($\alpha = 0, \beta = 1.0, \gamma = 0.1$), as reported by the authors this is the one with the best performance. The capacity of the discriminator networks has been partially reduced to fit the available hardware and to speedup the training, this did not affected significantly the results. We used Adam optimizer with learning rate $10^{-4}$ as in the original paper.

**SimCLR (Chen et al., 2020)** The code has been adapted from the implementation provided by the authors[4] and other open-source implementations (see code for details). To have a fair comparison with our method we used the same MLP head, the same data augmentation strategy, and optimizer (Adam with learning rate $10^{-3}$). We used a temperature of 0.5 in all the experiments, this was reported as the consistent optimal value regardless of the batch sizes in the original paper. We could not replicate the original setup reported by the authors on very large mini-batches, since it is computationally expensive, requiring 32 to 128 cores on Tensor Processing Units (TPUs). We adapted the setup to our available hardware (described in Appendix A.4), and we guaranteed a fair comparison by using a comparable number of pairs. In particular, we used a mini-batch of 128 images, which results in $6.5 \times 10^4$ pairs, this is similar (or superior) to the number of pairs compared by our method which is $6.3 \times 10^4$ for $K = 32$, $3.8 \times 10^4$ for $K = 25$, and $1.5 \times 10^4$ for $K = 16$.

## A.6    Finetuning experiments

All methods are trained for 300 epochs on the unlabeled portion of the STL-10 dataset, using the same hyperparameters and augmentations described before and a ResNet34 backbone. In the finetuning stage the pretrained backbone is coupled with a linear classifier and both of them are trained using Adam optimizer for 100 epochs with mini-batch of size 32. A lower learning rate for the backbone ($10^{-4}$) respect to the linear classifier ($10^{-3}$) has been used. Both learning rates are divided by 10 at 50% and 75% of total epochs. The same augmentations of Ji et al. (2019) have been used for the finetuning stage. Those consists of affine transformations (50% chance) sampled from a uniform distribution $[min, max]$: random rotation ($min = -18°, max = 18°$), scale ($min = 0.9, max = 1.1$), translation ($min = 0, max = 0.1$), shear ($min = -10, max = 10$), and bilinear interpolation, cutout (50% chance) with $32 \times 32$ patches. Same schedule and augmentations have also been used to train (from scratch) the supervised baseline on the labeled set of data (100 epochs).

## A.7    Semi-supervised experiments

We adapted our method to the semi-supervised case by coupling instances sampled from the same category. Those instances represented a portion of the total number of pairs in the mini-batch depending on the percentage of available labels. Results for each conditions are the average of three seeds. We used the same hyperparameters described in the linear evaluation phase. We did not prevent possible collisions between classes during the allocation of negative pairs. Collisions are unlikely in datasets with medium/high number of classes, but a slight performance improvement could be obtained if negatives are paired without collisions.

## A.8    Qualitative analysis experiments

For the qualitative analysis we compared the representations generated by the supervised baseline, Rotation Net, and our method on CIFAR-10 with a ResNet-32 backbone at the end of the training (200 epochs). The query images were randomly sampled and the representations compared using Euclidean distance. For the t-SNE analysis we used the Scikit implementation of the algorithms and set the hyperparameters as follows: 1000 iterations, Euclidean metric, random init, perplexity 30, learning rate 200.

# B Additional results

## B.1 Linear evaluation

Table 2: Linear evaluation. Self-supervised training on unlabeled data and linear evaluation on labeled data. Comparison between three datasets (CIFAR-10, CIFAR-100, tiny-ImageNet) for a shallow (Conv-4) and a deep (ResNet-32) backbone. Mean accuracy (percentage) and standard deviation over three runs. Best results highlighted in bold.

| | Conv-4 | | | ResNet-32 | | |
|---|---|---|---|---|---|---|
| Method | CIFAR-10 | CIFAR-100 | tiny-ImageNet | CIFAR-10 | CIFAR-100 | tiny-ImageNet |
| Supervised (upper bound) | 80.46±0.39 | 49.29±0.85 | 36.47±0.36 | 90.87±0.41 | 65.32±0.22 | 50.09±0.32 |
| Random Weights (lower bound) | 32.92±1.88 | 10.79±0.59 | 6.19±0.13 | 27.47±0.83 | 7.65±0.44 | 3.24±0.43 |
| DeepCluster (Caron et al., 2018) | 42.88±0.21 | 21.03±1.56 | 12.60±1.23 | 43.31±0.62 | 20.44±0.80 | 11.64±0.21 |
| RotationNet (Gidaris et al., 2018) | 56.73±1.71 | 27.45±0.80 | 18.40±0.95 | 62.00±0.79 | 29.02±0.18 | 14.73±0.48 |
| Deep InfoMax (Hjelm et al., 2019) | 44.60±0.27 | 22.74±0.21 | 14.19±0.13 | 47.13±0.45 | 24.07±0.05 | 17.51±0.15 |
| SimCLR (Chen et al., 2020) | 60.43±0.26 | 30.45±0.41 | 20.90±0.15 | **77.02±0.64** | 42.13±0.35 | 25.79±0.40 |
| *Relational Reasoning* (ours) | **61.03±0.23** | **33.38±1.02** | **22.31±0.19** | 74.99±0.07 | **46.17±0.16** | **30.54±0.42** |

Table 3: Linear evaluation on SlimageNet (Antoniou et al., 2020). This dataset is more challenging than ImageNet, since it only has 160 low-resolution ($64 \times 64$) color images for each one of the 1000 classes of ImageNet. Below is reported the linear evaluation accuracy on labeled data with a ResNet-32 backbone, after training on unlabeled data. Mean accuracy (percentage) and standard deviation over three runs. Best result highlighted in bold.

| Method | SlimageNet |
|---|---|
| Supervised (upper bound) | 33.94±0.21 |
| Random Weights (lower bound) | 0.79±0.09 |
| RotationNet (Gidaris et al., 2018) | 7.25±0.28 |
| SimCLR (Chen et al., 2020) | 14.32±0.24 |
| *Relational Reasoning* (ours) | **15.81±0.72** |

## B.2 Domain transfer

Table 4: Domain transfer. Training with self-supervision on unlabeled CIFAR-10 linear evaluation on CIFAR-100 ($10 \rightarrow 100$), and viceversa ($100 \rightarrow 10$). $\Delta$ indicates the difference between the accuracy in the standard setting (unsupervised train and linear evaluation on the same dataset) and the accuracy in the transfer setting (unsupervised train on first dataset and linear evaluation on the second dataset). Mean accuracy (percentage) and standard deviation over three runs. Best results highlighted in bold.

| | Conv-4 | | | | ResNet-32 | | | |
|---|---|---|---|---|---|---|---|---|
| Method | $10 \rightarrow 100$ | $\Delta$ | $100 \rightarrow 10$ | $\Delta$ | $10 \rightarrow 100$ | $\Delta$ | $100 \rightarrow 10$ | $\Delta$ |
| Supervised (upper bound) | 32.06±0.63 | -17.23 | 64.00±1.07 | -16.46 | 33.98±0.70 | -31.34 | 71.01±0.44 | -19.86 |
| Random Weights (lower bound) | 10.79±0.59 | n/a | 32.92±1.89 | n/a | 7.65±0.44 | n/a | 27.47±0.83 | n/a |
| DeepCluster (Caron et al., 2018) | 19.68±1.23 | -1.35 | 43.59±1.31 | +0.71 | 18.37±0.41 | -2.07 | 43.39±1.84 | +0.08 |
| RotationNet (Gidaris et al., 2018) | 26.06±0.09 | -1.39 | 51.86±0.36 | -4.87 | 27.02±0.20 | -2.00 | 52.22±0.70 | -9.78 |
| Deep InfoMax (Hjelm et al., 2019) | 22.35±0.12 | -0.39 | 43.30±0.15 | -1.30 | 23.73±0.04 | -0.34 | 45.05±0.24 | -2.08 |
| SimCLR (Chen et al., 2020) | 29.20±0.08 | -1.25 | 54.73±0.60 | -5.70 | 36.21±0.16 | -5.92 | 65.59±0.76 | -11.43 |
| *Relational Reasoning* (ours) | **31.84±0.23** | -1.54 | **57.30±0.26** | -3.73 | **41.50±0.35** | -4.67 | **67.81±0.42** | -7.18 |

## B.3 Grain

Table 5: Grain. Training with self-supervision on unlabeled CIFAR-100, and linear evaluation on labeled CIFAR-100 Fine-Grained (100 classes) and CIFAR-100-20 Coarse-Grained (20 super-classes). Mean accuracy (percentage) and standard deviation over three runs. Best results highlighted in bold.

| | Conv-4 | | ResNet-32 | |
|---|---|---|---|---|
| **Method** | **Fine-Grain** | **Coarse-Grain** | **Fine-Grain** | **Coarse-Grain** |
| Supervised (upper bound) | 49.29±0.85 | 59.91±0.62 | 65.32±0.22 | 76.35±0.57 |
| Random Weights (lower bound) | 10.79±0.59 | 19.94±0.31 | 7.65±0.44 | 16.56±0.48 |
| DeepCluster (Caron et al., 2018) | 21.03±1.56 | 30.07±2.06 | 20.44±0.80 | 29.49±1.36 |
| RotationNet (Gidaris et al., 2018) | 27.45±0.80 | 35.49±0.17 | 29.02±0.19 | 40.45±0.39 |
| Deep InfoMax (Hjelm et al., 2019) | 22.74±0.21 | 32.36±0.43 | 24.07±0.05 | 33.92±0.34 |
| SimCLR (Chen et al., 2020) | 30.45±0.41 | 37.72±0.14 | 42.13±0.35 | 51.88±0.48 |
| *Relational Reasoning* (ours) | **33.38±1.02** | **40.86±1.03** | **46.17±0.17** | **52.44±0.47** |

## B.4 Finetuning

Table 6: Finetuning. Comparison with other results reported in the literature on unsupervised training and finetuning on the STL-10 dataset. Best result in bold. *Local* refers to our local reproduction of the method, with results reported as *best (mean ± std)* on three runs with different seeds. Note that backbone and learning schedule may differ. The ResNet-34 backbone is much larger than ResNet-32 ($21.3 \times 10^6$ vs $0.47 \times 10^6$), showing that the proposed method can be effectively scaled.

| **Method** | **Reference** | **Backbone** | **Accuracy** |
|---|---|---|---|
| Supervised (crop + cutout) | DeVries and Taylor (2017) | WideResnet-16-8 | 87.30 |
| Supervised (scattering) | Oyallon et al. (2017) | Hybrid-WideResnet | 87.60 |
| Exemplars (Dosovitskiy et al., 2014) | Dosovitskiy et al. (2014) | Conv-3 | 72.80 |
| Artifacts (Jenni and Favaro, 2018) | Jenni and Favaro (2018) | Custom | 80.10 |
| ADC (Haeusser et al., 2018) | Ji et al. (2019) | ResNet-34 | 56.70 |
| DeepCluster (Caron et al., 2018) | Ji et al. (2019) | ResNet-34 | 73.40 |
| Deep InfoMax (Hjelm et al., 2019) | Ji et al. (2019) | AlexNet | 77.00 |
| Invariant Info Clustering (Ji et al., 2019) | Ji et al. (2019) | ResNet-34 | 88.80 |
| Supervised (affine + cutout) | Local | ResNet-34 | 72.04 (69.82 ± 3.36) |
| DeepCluster (Caron et al., 2018) | Local | ResNet-34 | 74.00 (73.37 ± 0.55) |
| RotationNet (Gidaris et al., 2018) | Local | ResNet-34 | 83.77 (83.29 ± 0.44) |
| Deep InfoMax (Hjelm et al., 2019) | Local | ResNet-34 | 76.45 (76.03 ± 0.37) |
| SimCLR (Chen et al., 2020) | Local | ResNet-34 | 89.44 (89.31 ± 0.14) |
| *Relational Reasoning* (ours) | Local | ResNet-34 | **90.04 (89.67 ± 0.33)** |

## B.5 Performance with different backbones

Table 7: Comparison on different backbones: linear evaluation. Comparison between four backbones of different depth for baselines and the three best performing methods. Training with self-supervision on unlabeled CIFAR-10 and CIFAR-100, and linear evaluation on labeled version of the same datasets. Mean accuracy (percentage) and standard deviation over three runs. Best results highlighted in bold.

| | Conv-4 | | ResNet-8 | | ResNet-32 | | ResNet-56 | |
|---|---|---|---|---|---|---|---|---|
| **Method** | **CIFAR-10** | **CIFAR-100** | **CIFAR-10** | **CIFAR-100** | **CIFAR-10** | **CIFAR-100** | **CIFAR-10** | **CIFAR-100** |
| Supervised (upper bound) | 80.46±0.39 | 49.29±0.85 | 87.08±0.17 | 59.41±1.15 | 90.87±0.41 | 65.32±0.22 | 91.40±0.30 | 67.54±0.32 |
| Random Weights (lower bound) | 32.92±1.89 | 10.79±0.59 | 35.94±1.39 | 13.08±0.91 | 27.47±0.83 | 7.65±0.44 | 13.53±3.66 | 1.88±0.14 |
| RotationNet (Gidaris et al., 2018) | 56.73±1.71 | 27.45±0.80 | 62.73±0.94 | 32.09±0.87 | 62.00±0.79 | 29.02±0.18 | 61.66±1.11 | 28.24±0.23 |
| SimCLR (Chen et al., 2020) | 60.43±0.26 | 30.45±0.41 | **69.85±0.58** | 36.23±0.15 | **77.02±0.64** | 42.13±0.35 | **78.75±0.24** | 44.33±0.48 |
| *Relational Reasoning* (ours) | **61.03±0.23** | **33.38±1.02** | 67.97±0.58 | **38.18±0.63** | 74.99±0.07 | **46.17±0.17** | 77.51±0.00 | **47.90±0.27** |

Table 8: Comparison on different backbones: grain. Comparison between four backbones of different depth for baselines and the three best performing methods. Training with self-supervision on unlabeled CIFAR-100 and linear evaluation on labeled version of the same datasets with 100 labels (fine) or 20 super-labels (coarse). Mean accuracy (percentage) and standard deviation over three runs. Best results highlighted in bold.

| Method | Conv-4 | | ResNet-8 | | ResNet-32 | | ResNet-56 | |
|---|---|---|---|---|---|---|---|---|
| | Fine | Coarse | Fine | Coarse | Fine | Coarse | Fine | Coarse |
| Supervised (upper bound) | 49.29±0.85 | 59.91±0.62 | 59.41±1.15 | 70.12±0.33 | 65.32±0.22 | 76.35±0.57 | 67.54±0.32 | 77.60±0.43 |
| Random Weights (lower bound) | 10.79±0.59 | 19.94±0.31 | 13.08±0.91 | 23.12±0.90 | 7.65±0.44 | 16.56±0.48 | 1.88±0.14 | 6.88±0.35 |
| RotationNet (Gidaris et al., 2018) | 27.45±0.80 | 35.49±0.17 | 32.09±0.87 | 41.21±0.94 | 29.02±0.18 | 40.45±0.39 | 28.24±0.23 | 39.16±0.35 |
| SimCLR (Chen et al., 2020) | 30.45±0.41 | 37.72±0.14 | 36.23±0.15 | 43.78±0.92 | 42.13±0.35 | 51.87±0.48 | 44.33±0.48 | 54.09±0.15 |
| *Relational Reasoning* (ours) | **33.38±1.02** | **40.86±1.03** | **38.18±0.63** | **45.36±0.55** | **46.17±0.17** | **52.44±0.47** | **47.90±0.27** | **54.90±0.07** |

Table 9: Comparison on different backbones: domain transfer. Comparison between four backbones of different depth for baselines and the three best performing methods. Training with self-supervision on unlabeled CIFAR-10 linear evaluation on CIFAR-100 ($10 \rightarrow 100$), and viceversa ($100 \rightarrow 10$). Mean accuracy (percentage) and standard deviation over three runs. Best results highlighted in bold.

| Method | Conv-4 | | ResNet-8 | | ResNet-32 | | ResNet-56 | |
|---|---|---|---|---|---|---|---|---|
| | $10 \rightarrow 100$ | $100 \rightarrow 10$ | $10 \rightarrow 100$ | $100 \rightarrow 10$ | $10 \rightarrow 100$ | $100 \rightarrow 10$ | $10 \rightarrow 100$ | $100 \rightarrow 10$ |
| Supervised (upper bound) | 32.06±0.63 | 64.00±1.07 | 36.83±0.36 | 71.20±0.18 | 33.98±0.70 | 71.01±0.44 | 33.92±0.50 | 71.97±0.17 |
| Random Weights (lower bound) | 10.79±0.59 | 32.92±1.89 | 13.08±0.91 | 35.94±1.39 | 7.65±0.44 | 27.47±0.83 | 1.88±0.14 | 13.53±3.66 |
| RotationNet (Gidaris et al., 2018) | 26.06±0.09 | 51.86±0.36 | 31.60±0.54 | 56.85±0.13 | 27.02±0.20 | 52.22±0.70 | 27.25±0.62 | 51.82±0.58 |
| SimCLR (Chen et al., 2020) | 29.20±0.08 | 54.73±0.60 | 34.46±0.78 | 61.34±0.24 | 36.21±0.16 | 65.59±0.76 | 36.79±0.45 | 66.19±0.80 |
| *Relational Reasoning* (ours) | **31.84±0.23** | **57.30±0.26** | **36.07±0.35** | **63.24±0.52** | **41.50±0.35** | **67.81±0.42** | **42.19±0.28** | **68.66±0.21** |

## B.6 Number of augmentations

Table 10: Accuracy with respect to the number of augmentations $K$. Methods have been trained on CIFAR-10 with a Conv-4 backbone for 100 epochs. The input mini-batch has been augmented $K$ times then given as input. Results are the average accuracy (linear evaluation) of three runs on the validation set. Only the relational reasoning accuracy is positively correlated with $K$.

| Method | $K = 2$ | $K = 4$ | $K = 8$ | $K = 16$ | $K = 32$ |
|---|---|---|---|---|---|
| Supervised | 79.61±0.47 | 79.76±0.54 | 79.96±0.71 | 79.56±0.49 | 80.00±0.45 |
| RotationNet (Gidaris et al., 2018) | 51.58±0.49 | 51.51±1.02 | 52.62±0.68 | 52.85±1.24 | 52.25±1.06 |
| *Relational Reasoning* (ours) | 55.31±0.58 | 58.05±0.67 | 59.24±0.51 | 60.26±0.59 | 60.33±0.36 |

## B.7 Semi-supervised and supervised

Table 11: Test accuracy on *CIFAR-10* with respect to the percentage of labeled data available. Methods have been trained with a ResNet-32 backbone (200 epochs), followed by linear evaluation on the entire labeled dataset (100 epochs). The quality of the representations improves with the number of labeled data available.

| Method | 0% | 1% | 10% | 25% | 50% | 100% |
|---|---|---|---|---|---|---|
| Supervised | n/a | n/a | n/a | n/a | n/a | 90.87±0.41 |
| *Relational Reasoning* (ours) | 74.99±0.07 | 76.55±0.27 | 80.14±0.35 | 85.30±0.28 | 89.35±0.11 | 90.66±0.23 |

Table 12: Test accuracy on *CIFAR-100* with respect to the percentage of labeled data available. Methods have been trained with a ResNet-32 backbone (200 epochs), followed by linear evaluation on the entire labeled dataset (100 epochs). The quality of the representations improves with the number of labeled data available.

| Method | 0% | 1% | 10% | 25% | 50% | 100% |
|---|---|---|---|---|---|---|
| Supervised | n/a | n/a | n/a | n/a | n/a | 65.32±0.22 |
| *Relational Reasoning* (ours) | 46.17±0.17 | 46.10±0.29 | 49.55±0.36 | 54.44±0.58 | 58.52±0.70 | 58.96±0.28 |

## B.8 Ablations

Table 13: Ablation of the aggregation function. Training with relational self-supervision on unlabeled CIFAR-10 and CIFAR-100, and linear evaluation on labeled datasets (Conv-4). Mean accuracy (percentage) and standard deviation over three runs on a validation set (obtained sampling 20% of the images from the training set). Best results highlighted in bold.

| Aggregation | Analytical form | CIFAR-10 | CIFAR-100 |
|---|---|---|---|
| Sum | $a_{\text{sum}}(\mathbf{z}_i, \mathbf{z}_j) = \mathbf{z}_i + \mathbf{z}_j$ | 57.60±0.23 | 29.45±0.69 |
| Mean | $a_{\text{mean}}(\mathbf{z}_i, \mathbf{z}_j) = \frac{\mathbf{z}_i + \mathbf{z}_j}{2}$ | 57.77±0.74 | 29.15±0.80 |
| Maximum | $a_{\text{max}}(\mathbf{z}_i, \mathbf{z}_j) = \max(\mathbf{z}_i, \mathbf{z}_j)$ | 56.45±1.15 | 26.58±1.26 |
| Concatenation | $a_{\text{cat}}(\mathbf{z}_i, \mathbf{z}_j) = \mathbf{z}_i \frown \mathbf{z}_j$ | **60.81±0.25** | **32.36±0.73** |

Table 14: Ablation of the relation head. The models have been trained on unlabeled CIFAR-10 and CIFAR-100 and tested on various benchmarks with a ResNet32 backbone for 200 epochs (mean accuracy and standard deviation of 3 runs). We consider three head types: *(a) dot product* between the pairs encoded through the backbone, followed by BCE loss; *(b) Encoder + dot product*, aggregation is not performed, for each encoded representation an MLP performs a second encoding, then dot product is applied between pairs and the BCE loss minimized (similar to SimCLR, Chen et al. 2020); *(c) Relation module* corresponds to the proposed method where encodings are aggregated (concatenation) and passed through an MLP for binary classification. All the other factors are kept constant for a fair comparison (e.g. augmentation strategy, mini-batch size). Best results in bold.

| Head type | Linear Evaluation | | Domain Transfer | | Grain |
|---|---|---|---|---|---|
| | CIFAR-10 | CIFAR-100 | 10→100 | 100→10 | CIFAR-100-20 |
| (a) dot product | 72.74±0.22 | 28.77±0.44 | 18.19±0.10 | 51.9±0.50 | 45.05±1.07 |
| (b) Encoder + dot product | 59.44±0.59 | 29.91±1.28 | 28.29±0.90 | 53.65±0.85 | 36.94±1.30 |
| (c) *Relation module* (ours) | **74.99±0.07** | **46.17±0.17** | **41.50±0.35** | **67.81±0.42** | **52.44±0.47** |

Figure 3: Ablation of the relation head (graphical illustration). Comparison between the two ablations in (a) and (b), and the full model in (c). In (a) the head is removed and the dot product $\langle z_1, z_2 \rangle$ is used to compare the representations pair. In (b) the relation head is replaced with an encoder $g_\phi$ that projects each representation in another latent space where the dot product is performed. In (c) is showed the full model, with the relation module $r_\phi$ taking in input the aggregated pair. In all cases is minimized the binary cross-entropy loss (BCE) over positive and negative pairs.

(a) Relational Reasoning (ours)                    (b) RotationNet

Figure 4: Image retrieval given 25 random queries (not cherry-picked) on CIFAR-10 with ResNet-32. The query is the leftmost image (red frame), followed by the top-10 most similar images (Euclidean distance) in representation space. Comparison between (a) self-supervised relational reasoning (ours), and (b) self-supervised rotation prediction (Gidaris et al., 2018). Our method shows better distinction between categories which are hard to separate, (e.g. ships vs planes in row 4, trucks vs cars in row 12). Moreover, the lower sample variance and the higher similarity with the query, indicates a fine-grained organization in representation space (e.g. red sport cars in row 1, long white trucks in rows 11 and 12, deer with snow in row 16, blue car in row 22, dog breeds in row 25).

## B.10 Image retrieval: error analysis

Figure 5: Confusion matrix obtained sampling 500 images per class (CIFAR-10, ResNet-32) and retrieving the top-10/100/1000 (top/middle/bottom table) closest images in representation space via Euclidean distance. Accuracy in percentage (three seeds) over correct retrievals (same category). Comparison between (a) self-supervised relational reasoning (ours), and (b) self-supervised rotation prediction (Gidaris et al., 2018). The proposed method shows a superior accuracy across all categories while being more robust against misclassification errors.

## B.11 Representations: qualitative analysis

(a) Supervised   (b) Relational Reasoning (ours)   (c) RotationNet

Figure 6: Visualization of t-SNE embeddings for the 10K test points in CIFAR-10. ResNet-32 backbone trained via (a) supervised learning, (b) self-supervised relational reasoning (ours), and (c) self-supervised rotation prediction (Gidaris et al., 2018). Our method shows a lower scattering, with clusters which are more distinct.

(a) Supervised   (b) Relational Reasoning (ours)   (c) RotationNet

Figure 7: Visualization of t-SNE embeddings for the 10K test points in CIFAR-10 divided in two super-categories: vehicles (plane, car, ship, truck), and animals (bird, cat, deer, dog, frog, horse). ResNet-32 backbone trained via (a) supervised learning, (b) self-supervised relational reasoning (ours), and (c) self-supervised rotation prediction (Gidaris et al., 2018). Our method shows a better split, lower scattering, and a minor overlap between the two super-categories.

## C  Pseudo-code of the method

---

**Algorithm 1** Self-supervised relational learning: training function and shuffling without collisions.

---

**Require:** $\mathcal{D} = \{\mathbf{x}_n\}_{n=1}^N$ unlabeled training set; $\mathcal{A}(\cdot)$ augmentation distribution; $\boldsymbol{\theta}$ parameters of $f_\theta$ (neural network backbone); $\boldsymbol{\phi}$ parameters of $r_\phi$ (relation module); aggregation function $a(\cdot, \cdot)$; $\alpha$ and $\beta$ learning rate hyperparameters; $K$ number of augmentations; $M$ mini-batch size;

1: **function** TRAIN($\mathcal{D}, \alpha, \beta, M, K, \boldsymbol{\theta}, \boldsymbol{\phi}$)
2:     **while** not done **do**
3:         $\mathcal{B} = \{\mathbf{x}_m\}_{m=1}^M \sim \mathcal{D}$                                                 ▷ Sampling a mini-batch
4:         **for** $k = 1$ to $K$ **do**
5:             $\mathcal{B}^{(k)} \sim \mathcal{A}(\mathcal{B})$                                 ▷ Sampling $K$ mini-batch augmentations
6:             $\mathcal{Z}^{(k)} = f_\theta(\mathcal{B}^{(k)})$                           ▷ Forward pass in the backbone
7:         **end for**
8:         $\mathcal{P} = \{\}$                           ▷ Empty set to store aggregated pairs and targets
9:         **for** $i = 1$ to $K - 1$ **do**
10:             **for** $j = i + 1$ to $K$ **do**
11:                 $\mathcal{P} \leftarrow \big(a(\mathcal{Z}^{(i)}, \mathcal{Z}^{(j)}), \mathbf{t} = \mathbf{1}\big)$        ▷ Aggregating and appending positive pairs
12:                 $\tilde{\mathcal{Z}}^{(j)} = \text{SHUFFLE}(\mathcal{Z}^{(j)})$            ▷ Shuffling without collisions
13:                 $\mathcal{P} \leftarrow \big(a(\mathcal{Z}^{(i)}, \tilde{\mathcal{Z}}^{(j)}), \mathbf{t} = \mathbf{0}\big)$        ▷ Aggregating and appending negative pairs
14:             **end for**
15:         **end for**
16:         $\mathbf{y} = r_\phi(\mathcal{P})$                           ▷ Forward pass in the relation module
17:         $\mathcal{L} = \text{BCE}(\mathbf{y}, \mathbf{t})$                      ▷ Estimating the Binary Cross-Entropy loss
18:         $\boldsymbol{\theta} \leftarrow \boldsymbol{\theta} - \alpha\nabla_\theta\mathcal{L}$                      ▷ Updating backbone
19:         $\boldsymbol{\phi} \leftarrow \boldsymbol{\phi} - \beta\nabla_\phi\mathcal{L}$                    ▷ Updating relation module
20:     **end while**
21:     **return** $\boldsymbol{\theta}, \boldsymbol{\phi}$                            ▷ Returning the learned weights
22: **end function**

23: **function** SHUFFLE($\mathcal{Z}$)
24:     $\tilde{\mathcal{Z}} = \mathcal{Z}$                               ▷ Copying the input set
25:     **for** $m = 1$ to $M$ **do**
26:         $\tilde{m} \sim \{1, \ldots, M\} \setminus \{m\}$               ▷ Sampling an index $\tilde{m} \neq m$
27:         $\tilde{\mathcal{Z}}_m \leftarrow \mathcal{Z}_{\tilde{m}}$         ▷ Assigning a random representation with index $\tilde{m}$
28:     **end for**
29:     **return** $\tilde{\mathcal{Z}}$                           ▷ Returning the shuffled set
30: **end function**

---

## D  Essential PyTorch code of the method

### D.1  Data loader

```python
import torchvision
from PIL import Image

class MultiCIFAR10(torchvision.datasets.CIFAR10):
  """Override torchvision CIFAR10 for multi-image management.
     Similar class can be defined for other datasets (e.g. CIFAR100).
  """
  def __init__(self, K, **kwds):
    super().__init__(**kwds)
    self.K = K # tot number of augmentations

  def __getitem__(self, index):
    img, target = self.data[index], self.targets[index]
    pic = Image.fromarray(img)
```

```
      img_list = list()
      if self.transform is not None:
        for _ in range(self.K):
          img_transformed = self.transform(pic.copy())
          img_list.append(img_transformed)
      else:
          img_list = img
      return img_list, target
```

## D.2 Augmentations

```
import torchvision.transforms as transforms

normalize = transforms.Normalize(mean=[0.491, 0.482, 0.447],
                                  std=[0.247, 0.243, 0.262]) # CIFAR10
color_jitter = transforms.ColorJitter(brightness=0.8, contrast=0.8,
                                       saturation=0.8, hue=0.2)
rnd_color_jitter = transforms.RandomApply([color_jitter], p=0.8)
rnd_gray = transforms.RandomGrayscale(p=0.2)
rnd_rcrop = transforms.RandomResizedCrop(size=32, scale=(0.08, 1.0),
                                         interpolation=2)
rnd_hflip = transforms.RandomHorizontalFlip(p=0.5)
train_transform = transforms.Compose([rnd_rcrop, rnd_hflip,
                                       rnd_color_jitter, rnd_gray,
                                       transforms.ToTensor(), normalize
                                       ])
```

## D.3 Self-supervised relational reasoning

```
import torch

class RelationalReasoning(torch.nn.Module):
  def __init__(self, backbone):
    super(RelationalReasoning, self).__init__()
    feature_size = 64*2 # multiply by 2 since aggregation='cat'
    self.backbone = backbone
    self.relation_head = torch.nn.Sequential(
                            nn.Linear(feature_size, 256),
                            nn.BatchNorm1d(256),
                            nn.LeakyReLU(),
                            nn.Linear(256, 1))

  def aggregate(self, features, K):
    relation_pairs_list = list()
    targets_list = list()
    size = int(features.shape[0] / K)
    shifts_counter=1
    for index_1 in range(0, size*K, size):
      for index_2 in range(index_1+size, size*K, size):
        # Using the 'cat' aggregation function by default
        pos_pair = torch.cat([features[index_1:index_1+size],
                              features[index_2:index_2+size]], 1)
        # Shuffle by rolling the mini-batch (negatives)
        neg_pair = torch.cat([
                   features[index_1:index_1+size],
                   torch.roll(features[index_2:index_2+size],
                   shifts=shifts_counter, dims=0)], 1)
        relation_pairs_list.append(pos_pair)
        relation_pairs_list.append(neg_pair)
        targets_list.append(torch.ones(size, dtype=torch.float32))
        targets_list.append(torch.zeros(size, dtype=torch.float32))
        shifts_counter+=1
        if(shifts_counter>=size):
            shifts_counter=1 # avoid identity pairs
    relation_pairs = torch.cat(relation_pairs_list, 0)
```

```
      targets = torch.cat(targets_list, 0)
      return relation_pairs, targets

  def train(self, tot_epochs, train_loader):
    optimizer = torch.optim.Adam([
                  {'params': self.backbone.parameters()},
                  {'params': self.relation_head.parameters()}])
    BCE = torch.nn.BCEWithLogitsLoss()
    self.backbone.train()
    self.relation_head.train()
    for epoch in range(tot_epochs):
      # the real target is discarded (unsupervised)
      for i, (data_augmented, _) in enumerate(train_loader):
        K = len(data_augmented) # tot augmentations
        x = torch.cat(data_augmented, 0)
        optimizer.zero_grad()
        # forward pass (backbone)
        features = self.backbone(x)
        # aggregation function
        relation_pairs, targets = self.aggregate(features, K)
        # forward pass (relation head)
        score = self.relation_head(relation_pairs).squeeze()
        # cross-entropy loss and backward
        loss = BCE(score, targets)
        loss.backward()
        optimizer.step()
        # estimate the accuracy
        predicted = torch.round(torch.sigmoid(score))
        correct = predicted.eq(targets.view_as(predicted)).sum()
        accuracy = (100.0 * correct / float(len(targets)))

        if(i%100==0):
          print('Epoch [{}][{}/{}] loss: {:.5f}; accuracy: {:.2f}%' \
            .format(epoch+1, i+1, len(train_loader)+1,
                    loss.item(), accuracy.item()))
```

## D.4 Main

```
def main():
  backbone = Conv4() # it should be a CNN with 64 linear output units
  model = RelationalReasoning(backbone)
  train_set = MultiCIFAR10(K=4, # it should be K=32 for CIFAR10/100
                           root='data', train=True,
                           transform=train_transform,
                           download=True)
  train_loader = torch.utils.data.DataLoader(train_set,
                                             batch_size=64,
                                             shuffle=True)
  model.train(tot_epochs=200, train_loader=train_loader)
  torch.save(model.backbone.state_dict(), './backbone.tar')
```

## Footnotes

[2]`https://github.com/facebookresearch/deepcluster`

[3]`https://github.com/rdevon/DIM`

[4]`https://github.com/google-research/simclr`