[Reviews · NeurIPS 2020]

Review 1

Summary and Contributions: ==== Post rebuttal update ==== I empathize with the author's argument of not having enough compute to do self-supervised pre-training of a ResNet50 on ImageNet. If the compute does become available somehow, please do include these numbers in the final manuscript. The authors discuss a lack of standardization in the self-supervision community with regards to evaluation protocols. In the community's defense, when AlexNet was still the go to architecture for self-supervision papers, the following protocol was used: ImageNet linear eval, Places 205 linear eval, Fast(er)-RCNN VOC 2007 detection fine-tuning, VOC 2007 multiclass classification fine-tuning, VOC 2012 + train-aug FCN semantic segmentation fine-tuning. Learning rate schedules, data augmentation etc. were fixed. Things are now in a flux with regards to self-supervision using ResNets, but standardization was attempted by https://arxiv.org/abs/1905.01235 . The authors answered several of our concerns. My overall score remains positive. ======== End of post rebuttal update =========== This paper presents a novel formulation for self-supervised representation learning for downstream recognition tasks. The pretext task is constructed in a manner very similar to modern contrastive representation learning methods. In more detail: The data samples are augmented. Different augmentations of the same input are treated as related and augmentations of different inputs are treated as unrelated. The loss is meant to classify an aggregation of related embeddings as label 1 (positive), and unrelated embeddings as label 0 (negative). The paper demonstrates linear evaluation and transfer learning performance across a range of downstream small to medium sized datasets: Cifar-10, Cifar-100, Tiny-ImgNet, STL-10, etc.

Strengths: The way this paper implements their idea, they have O(M K**2) pairs to classify as related/unrelated. M is the batch-size and K is the number of data augmentations. Thus the dependence on batch-size is linear and not quadratic as in SimCLR. Further they are able to demonstrate significant gains from using multiple augmentations per mini-batch. I believe these difference with contrastive representation learning are relevant to the NeurIPS community. The evaluation is thorough in terms of using multiple depths of the backbone, various datasets, and various downstream scenarios. Their method also shows impressive results on all these evaluations.

Weaknesses: While the evaluation is very extensive, it is restricted to image classification problems and to small/medium sized benchmarks (considering both image-size and dataset-size). The authors attempt to standardize the self-supervised evaluation pipeline in the visual domain, but miss several key vision tasks such as multi-class classification, object detection, and semantic segmentation. These tasks were used extensively in the evaluation of self-supervised backbones by prior work [Zhang et al. 2016, Zhang et al. 2017, Noorozi and Favaro 2016]. These were recently revised and standardized in the ResNet world by [Goyal et al., arxiv 2019]. A variation of these was also used in [He et.al., 2019] - a state-of-the-art contrastive learning method. The paper can further its impact by evaluating on large scale benchmarks such as: ImageNet linear evaluation (although this has been criticized as overly sensitive to the learning rate schedule) ImageNet 1%, 10% fine-tuning Pascal VOC Object Detection (see section 4.2.1 of He et.al. 2019 - https://openaccess.thecvf.com/content_CVPR_2020/papers/He_Momentum_Contrast_for_Unsupervised_Visual_Representation_Learning_CVPR_2020_paper.pdf ) The above are not expected for the rebuttal. References ========= [Goyal et al., arxiv 2019] Scaling and Benchmarking Self-Supervised Visual Representation Learning, Priya Goyal, Dhruv Mahajan, Abhinav Gupta, Ishan Misra, ArXiv: https://arxiv.org/abs/1905.01235

Correctness: The aggregator function is described as commutative. However, most of the results use concatination which is non-commutative. Is commutativity not important? When using concatination does the loss sample both (z(i), z(j)) and (z(j), z(i)) [lines 181-182]?

Clarity: The paper is well written and easy to understand. The method is well motivated.

Relation to Prior Work: Yes, especially the relationship with contrastive representation learning is elaborated in the discussions section.

Reproducibility: Yes

Additional Feedback: Typo in line 148: f_\theta instead of f_\phi.


Review 2

Summary and Contributions: This paper propose a relational reasoning based self-supervised learning framework. The model improves the recent contrastive learning model, e.g., SimCLR, MoCo, by changing the similarity computation from simple cosine similarity to a relation module. The authors show extensive experiments to demonstrate the effectiveness of the model.

Strengths: 1. The model propose a self-supervised learning framework which achieve the SOTA performance compare to recent slef-supervised learning methods. 2. The idea of modeling the self-supervised learning into relational reasoning is clear. 3. Extensive experiments and ablation study are conducted in a fair and standard setting to compare with different methods.

Weaknesses: Experiments are conducted on ResNet34 or ResNet36, as we can see in previous self-supervied learning works, e.g., SimCLR, MoCo, the self-supervised models may vary in performances with different backbones. So it would be better to show the effectiveness with more backbones.

Correctness: This paper propose a self-supervised learning framework based on relational reasoning which achieves state-of-the-art performance on several benchmark datasets. More detailed comparisons between different backbones would make the paper more convincing.

Clarity: Yes

Relation to Prior Work: Yes

Reproducibility: Yes

Additional Feedback:


Review 3

Summary and Contributions: ======================= Comments added after author feedback & discussion I think the authors did write a nice rebuttal and I am overall changing my rating to a (marginally) positive one. The authors did clarify some of my concerns about the experiments and I also empathize about the lack of compute to run experiments in a larger scale ie R50/Imagenet. However, the authors did not provide any extra ablation results regarding their novel points/changes and they actually metnion that this would be future work, ie the authors also do not really know which change helps how much: Multiple augmentations, the loss function and the way pairs are compared, are all changes that are mostly ablated together (there is an ablation for the latter in the supplementary but misses the most common dot product function), and it seems that the authors still dont have 1-by-1 ablations for those (over SimCLR). E.g. a version of the proposed approach, without the relation module and dot product as the function, ie dot product + BCE loss, would be a very interesting datapoint. What one could find (and this is very common I believe) is that most gains come from one or two of those, and I would love to understand which and how much. This paper would be a much stronger paper with such an ablation. Because of the above missing ablations, I am changing my recommendation to only a marginally positive one and I am urging the authors to actually include those in the next version of their manuscript (and not consider it "future work"). I also encourage them to run R50 experiments on imagenet, if they find the compute as this would ease comparisons to previous and future works. ======================= The authors present an approach for self-supervised learning (SSL) that is based on relational reasoning, in the sense that pairs of augmentations of the same image (positive pairs P) are to be differentiated from pairs of one augmentation of the same image with others from the mini-batch (negative pairs N). For creating positive pairs, they follow the augmentation of SimCLR (chen et al) but instead of one as in SimCLR, the create multiple augmentations. Unlike SimCLR that computes the dot product between P and N, here the authors propose to aggregate (ie concatenate in practice) them, and feed them to an MLP (relation module) that predicts whether they are matching or not with a focal BCE loss. They experiment on a number of small scale datasets and show that they can outperform the contrastive simCLR method as well as other .

Strengths: - SSL is a very active research area and a very important task to advance - The authors go "against" the recent trend to contrastive losses for SSL and use a BCE loss, showing it can also achieve strong performance; this is an interesting claim - The authors report results better than the recent contrastive learning method simCLR on small datasets

Weaknesses: A) In essence, the differences between simCLR and the proposed approach are not that many, and one could go progressively from the one to the other, exploring which change helps the most. See answer to the question "relation to prior work" below, for more discussion on this. This paper would be much stronger if it would guide us and let us know how performance increases with each change. B) Experiments lack on two major ways: a) are on smaller datasets and do not include eg the most common ImageNet-1k benchmark, that would enable the authors to directly compare to other recent related works; b) the are conducted with relatively small backbones and again do not include the common choice of resnet-50. The conv4 comparisons (and this is where most ablations are conducted) are to be taken with a pinch of salt as conv4 is a highly outdated architecture with lower learning capacity. C) Transfer learning experiments are only between small datasets and the domain transfer between CIFAR 10 and 100 is minimal. Apart from these experiments, for all other experiments the domain of the SSL training and the testing is exactly the same. This is a (very popular but) biased way of evaluating SSL and thats why all recent papers focus instead on true transfer learning performance, either across datasets of different domains or different tasks like object detection. D) In general, we know that SSL methods need a lot of hyperparameter tuning, something that was extensively done for the proposed approach but not for SimCLR. The latter was never evaluated on these datasets or more importantly these backbones. It is a weakness of the paper that it does not compare directly to any results from the SimCLR paper (which has some R50 CIFAR 10 experiments in their appendix that wouldn't be hard to compare). One

Correctness: The claims and method seem correct, the empirical methodology as well.

Clarity: The paper is very well written and easy to read and follow. Notation is clear.

Relation to Prior Work: If one disregards methods that were public as preprints but not yet officially presented at the time of the NeurIPS deadline (MoCo and MoCo-v2, CMC, CPC-v2) then the closest related works missing from comparison tables and discussions are (Wu et al) and (Zhuang et al). They are cited but neigher discussed not compared to. The differences to SimCLR, a method that the authors partially build upon -- they use their augmentation strategy, a very important aspect as recent related work has shown (see Tian et al 2020)--- are not clearly stated. In fact, if one uses the dot product as the aggregation function \alpha and a single augmentation, then the resulting approach would start getting very similar to simCLR. Taking one step back, I see 3 main differences between the two (and I would urge the authors to correct me in the rebuttal if needed): 1) the way pairs are created (proposed approach: multiple augmentations, randomly selected negatives - simCLR: one augmentation, exhaustively use all as negatives) 2) the way the pairs are compared (simCLR: dot product, proposed: concat and 2 layer MLP) 3) the loss function (BCE for each pair for the proposed, contrastive after a softmax over all pairs) It is worth noting here that a corollary of 2) is that pairs are created in the space of the transferable representations for the proposed approach and not in the loss space as in SimCLR (which further has an MLP before the dot product and after the transferable representations). Adding the above changes one by one would show which part helps most, something that is now unclear and a weakness of the paper. See the additional feedback below for some ablations.

Reproducibility: Yes

Additional Feedback: Some questions and clarifications: 1) Is the focal part of the loss important? What if \gamma = 1? did you ablate this parameter? 2) Is the accuracies reported for classifications the final accuracy on the val/test set (after a fixed number of X epochs) or the "best" accuracies during linear classifier learning? 3) In line 325 the authors claim that "we hypothesize that the difference in performance [to SimCLR] is mainly due to the loss function." But, as mentioned above, there are 3 main differences with simCLR and without the proper ablations this hypothesis is unfounded. Instead, it would be great to ablate each difference independently: eg a) use the pairs for the proposed approach (ie multiple augmentations, randomly selected negatives) with a contrastive loss b) use the focal BCE loss on top of SimCLR c) switch the order of "aggregation + MLP" to "MLP and dot product" for the proposed approach, and use the same BCE loss. My current rating reflect the fact that although this paper has some interesting ideas, a) the differences with contrastive learning methods are not clearly discussed b) the experimental validation are on small datasets and backbones and no results directly comparable to the SoTA. The gains over contrastive learning methods seem exargerated to me, because of these choices c) there are practically no transfer learning experiments beyond CIFAR 10<-> 100, two datasets with minimal domain shift. For all other results, the self-supervised representations are learned on the same dataset as the are tested on, a common by to my opinion flawed experimental setup. A Note on SSL papers in 2020: SSL has seen a surge of great new papers pushing the state of the art, and although most have been public before the NeurIPS deadline, they are only pre-prints. E.g. the state of the art method MoCo (He et al) had been out since late 2019 and cited by the proposed approach, and its followup (MoCo-v2) that mixes MoCo with SimCLR had been out since March. None of the two methods are compared to in the proposed manuscript, and the rules state they do not need to be. However, it would be better for the adoption of this work, to not only discuss but also compare with such methods that are already highly cited and the current state of the art.


Review 4

Summary and Contributions: The authors present a self-supervised representation learning method that trains a relational neural network to classify if a pair of embeddings are positive or negative. Positives embeddings come from augmentations of the same image while negatives come from augmentations of different images. The authors claim this setup results in better downstream task performance compared to existing SOTA self-supervised methods like RotNet and SimCLR.

Strengths: 1. The method is general and presents an interesting alternative to metric learning based contrastive loss. Instead of using the cosine similarity they use a learned similarity measure between two embeddings using fully connected layers on both of the embeddings. 2. While the comparison with SimCLR is not fair because of difference in experiments, the big claim is that this method of training leads to better performance on downstream tasks compared to SimCLR.

Weaknesses: 1. Missing comparison with MoCo (v1 and v2) [1, 2] and experiments on ImageNet. Since most methods compare extensively on ImageNet having some experiments on ImageNet would make a stronger case for the paper. 2. The idea of using fully-connected layers on top of the embeddings to learn a similarity measure has been explored before in the context of few-shot learning [3]. [1] He, K., Fan, H., Wu, Y., Xie, S., and Girshick, R. Momentum contrast for unsupervised visual representation learning. [2] Xinlei Chen, Haoqi Fan, Ross Girshick, Kaiming He. Improved Baselines with Momentum Contrastive Learning [3] Learning to Compare: Relation Network for Few-Shot Learning. Flood Sung, Yongxin Yang, Li Zhang, Tao Xiang, Philip H.S. Torr, Timothy M. Hospedales

Correctness: Explanations in "Discussion and conclusions" are not convincing: 1) Comparison with contrastive losses. "The larger set of positives allows relational reasoning to be more effective in disentangling fine-grained relations." The basis of this sentence is empirically comparing their method with SimCLR. The authors also say "We hypothesize that the difference in performance is mainly due to the loss function". Such a strong statement should only be made by running experiments using one algorithm. MoCo and SimCLR report the opposite finding that more negatives lead to better performance. 2) Why does cross-entropy work so well? "We argue that cross-entropy has been overlooked in the self-supervised learning literature, in favor of contrastive losses. Cross-entropy seems unable to provide a robust self-supervised signal since it is not a distance metric." It is unclear why the authors believe cross-entropy has been overlooked. Modern self-supervised methods (Jigsaw, RotNet, SimCLR, Local aggregation) all use cross-entropy losses. On the other hand, the key difference between their method and previous methods is the choice of the similarity function. While previous approaches use cosine similarity (or use a classification layer), the authors use a learned similarity function. It is not clear why authors say the cross-entropy has been overlooked.

Clarity: Paper is well written.

Relation to Prior Work: Well-written related work section.

Reproducibility: Yes

Additional Feedback: ***POST REBUTTAL UPDATE*** I thank the authors for their answers to my questions. I hope they can move some of our discussion on the loss to the main paper. Due to limited novelty of the approach I keep my score the same as before.

[Author Response · NeurIPS 2020]

**General considerations.** We thank the reviewers for their detailed comments and suggestions. We are glad that the
response has been mostly positive with the reviewers finding the method interesting and the empirical investigation
extensive. Before responding to each reviewer individually, we provide general considerations about experiments on
ImageNet with larger backbones, which has been identified as an issue by some of the reviewers. We will attempt to
get ResNet50/ImageNet runs for the final version. Recent self-supervised learning methods are expensive: SimCLR
(ResNet50, ImageNet) has been trained on 128-v3 TPUs cores, using a very large mini-batch of 4096 ($67.1 \times 10^6$ pairs).
A fair comparison of our method against this setup would require a mini-batch of 1024 and $K = 256$ augmentations
($66.8 \times 10^6$ pairs). To give an idea of the amount of computing required to run this experiment, consider that common
GeForce RTX 2080 or Titan-X are close to saturation on CIFAR-10 with ResNet32, mini-batch 64, $K = 32$ and
$6.3 \times 10^4$ pairs (two order of magnitude less than the desired target). To bypass this obstacle we provided extensive
experiments on ImageNet surrogates such as tiny-ImageNet (200 classes, 120K images) and SlimageNet (1000 classes,
200K images). We guaranteed a fair comparison with SimCLR by equalizing the number of pairs (see Section 5 of the
paper). Most importantly, we did not finetune our method; we used standard values for learning rates, focal loss, and
weight decay, showing that unlike other recent methods we do not need heavy-finetuning. We consider this setup a
reasonable compromise, as it prunes away inflated results and the need of large computational resources. Having said
that, we will try to get ImageNet runs by allocating more resources and engineering a highly-parallelizable solution.
However, it is unlikely that those resources will be enough for an extensive hyperparmeter search as for SimCLR. We
also stress the fact that the code will be released, giving the opportunity to compare it on larger benchmarks.

**Reviewer #1** marked the paper for clear acceptance (top 50% accepted papers), highlighting the efficiency of the
method, and the thorough empirical evaluation in terms of backbones, datasets, and downstream tasks. We thank the
reviewer for the positive feedback and the commentary, which has captured the essential strengths of this work. The
reviewer pointed out that the paper can further its impact by evaluating on different tasks and larger benchmarks, stating
that those experiments are not expected for the rebuttal. We have focused on classification and image retrieval, but we
agree on the importance of extending the evaluation to other tasks, this will be likely done in future work.

**Reviewer #2** recommends acceptance as slightly above threshold, highlighting the strong results, the clarity of the
method, the extensive empirical investigation and the fair experimental procedure. The reviewer pointed out that it may
be necessary to test the methods with larger backbones since self-supervised models may vary in performances with
different backbones. We agree with the reviewer, as explained above we did not run those experiments for computational
barriers. We will attempt a ResNet50 experiment on ImageNet for the final version.

**Reviewer #3** marked the paper as slightly below threshold but found the idea interesting, in particular the fact that a
binary cross-entropy loss can lead to strong results. We thank the reviewer for providing such a detailed commentary
and hope that the considerations below will resonate with the reviewer. *(1) Training procedure.* The reviewer states
that the procedure of training and testing on the same dataset is a popular but biased way of evaluating self-supervised
methods. As the reviewer correctly pointed out, this procedure (also known as "linear probes") has been widely adopted,
being used in a number of highly influential papers. This should be considered as a criticism to the way the community
is benchmarking methods rather than a criticism against our setup. We need (as a community) to define and validate
standardized protocols for self-supervised learning. Given the lack of such protocols, we have used what is common
choice to make our work comparable. *(2) Transfer learning experiments.* There is scarce agreement on appropriate
datasets and we consider this a corollary of the previous point: a general standardization issue. CIFAR 10/100 are a
good compromise since they have same image size, they are popular, and provide a realistic distribution shift. Models
trained on CIFAR10 must generalize to 90 unseen classes on CIFAR100 some of them remarkably different from
the base set (e.g. furniture, electrical devices, etc), our method excels in this setting. Additionally, note that we have
achieved SoTA with ResNet34 on STL-10 (see Table 1, appendix B.4), which has a domain shift between unlabeled
and labeled set. *(3) Clarifications.* Focal loss: there is a minor but consistent improvement from the use of focal loss
($+0.5\%$ on average). Evaluation: we reported the accuracy after a fixed amount of epochs (test set has never been
accessed during training). We also tried model selection through validation (held-out from training set); this did not
change the difference in accuracy between methods. *(4) Ablations.* The analysis of the reviewer about the three main
differences with SimCLR is correct. We will try those ablations in future work and add a clarification in the paper.

**Reviewer #4** recommends acceptance as above threshold, highlighting how the method is an interesting alternative
to contrastive approaches. *(1) Difference in performance due to the loss.* This was a speculative claim from our part,
we agree that it may not be the only factor at play. We can make it clear in the final version. *(2) Cross-entropy has
been overlooked.* We meant overlooked in the context of contrastive methods. NCE is a type of cross-entropy that
requires the estimation of an expensive partition function; this has been a default choice in SoTA methods (this is why:
"overlooked"). In our case, the use of a standard binary cross-entropy overcomes this issue. We can clarify this point for
the final version. *(3) Comparison with MoCo v1/v2.* We gave higher priority to SimCLR rather than MoCo-v1 (same
family but the former is stronger). MoCo-v2 was released in mid-March, we did not have time to implement it in our
framework (for the NeurIPS rules a comparison was not due).

[Meta-Review · NeurIPS 2020]

This paper presents a relational reasoning based self-supervised learning framework. All the reviewers like the paper and believe it could be impactful. There are concerns about experiments (not extensive, on small datasets, smaller backbone and only on classification). The authors provided a strong rebuttal addressing some of the concerns on experiments. Some ablation experiment concerns still remain. The AC agrees with all the reviewers that this is indeed a good paper and with more experiments (specifically on ablation and ImageNet) can become a really strong submission. AC committee recommends acceptance.